# BEHAVIOR-1K: A Benchmark for Embodied AI with 1,000 Everyday Activities and Realistic Simulation

**Chengshu Li**[*1], **Ruohan Zhang**[*1], **Josiah Wong**[*2], **Cem Gokmen**[*1],
**Sanjana Srivastava**[*1], **Roberto Martín-Martín**[*9,10], **Chen Wang**[*1], **Gabrael Levine**[*1],
**Michael Lingelbach**[3], **Jiankai Sun**[4], **Mona Anvari**[1], **Minjune Hwang**[1], **Manasi Sharma**[1],
**Arman Aydin**[1], **Dhruva Bansal**[1], **Samuel Hunter**[1], **Kyu-Young Kim**[1], **Alan Lou**[5],
**Caleb R Matthews**[1], **Ivan Villa-Renteria**[1], **Jerry Huayang Tang**[1], **Claire Tang**[1], **Fei Xia**[6],
**Silvio Savarese**[1,8,10], **Hyowon Gweon**[7,8], **C. Karen Liu**[1,8], **Jiajun Wu**[1,8], **Li Fei-Fei**[1,8]

Department of Computer Science[1], Department of Mechanical Engineering[2]
Neurosciences IDP[3], Department of Aeronautics and Astronautics[4]
Institute for Computational and Mathematical Engineering[5]
Department of Electrical Engineering[6], Department of Psychology[7]
Institute for Human-Centered Artificial Intelligence (HAI)[8]
Stanford University

The University of Texas at Austin[9], Salesforce Research[10]

**Abstract:** We present BEHAVIOR-1K, a comprehensive simulation benchmark for human-centered robotics. BEHAVIOR-1K includes two components, guided and motivated by the results of an extensive survey on '*what do you want robots to do for you?*'. The first is the definition of 1,000 everyday activities, grounded in 50 scenes (houses, gardens, restaurants, offices, etc.) with more than 5,000 objects annotated with rich physical and semantic properties. The second is OMNIGIBSON, a novel simulation environment that supports these activities via realistic physics simulation and rendering of rigid bodies, deformable bodies, and liquids. Our experiments indicate that the activities in BEHAVIOR-1K are long-horizon and dependent on complex manipulation skills, both of which remain a challenge for even state-of-the-art robot learning solutions. To calibrate the simulation-to-reality gap of BEHAVIOR-1K, we provide an initial study on transferring solutions learned with a mobile manipulator in a simulated apartment to its real-world counterpart. We hope that BEHAVIOR-1K's human-grounded nature, diversity, and realism make it valuable for embodied AI and robot learning research. Project website: https://behavior.stanford.edu.

**Keywords:** Embodied AI Benchmark, Everyday Activities, Mobile Manipulation

## 1 Introduction

Inspired by the progress that benchmarking brought to computer vision [1–11] and natural language processing [12–16], the robotics community has developed several benchmarks in simulation [17–30]. The broader goal of these benchmarks is to fuel the development of general, effective robots that bring major benefits to people's daily lives – human-centered AI that "serves human needs, goals, and values" [31–34]. Inspiring as they are, the tasks and activities in those benchmarks are designed by researchers; it remains unclear if they are addressing the actual needs of humans.

We observe that a human-centered robotic benchmark should not only be designed *for* human needs, but also originated *from* human needs: *what everyday activities do humans want robots to do for them?* To this end, we conduct an extensive survey with 1,461 participants (see Sec. 2) to rank a wide

---

\* indicates equal contribution
correspondence to {chengshu,zharu}@stanford.edu

6th Conference on Robot Learning (CoRL 2022), Auckland, New Zealand.

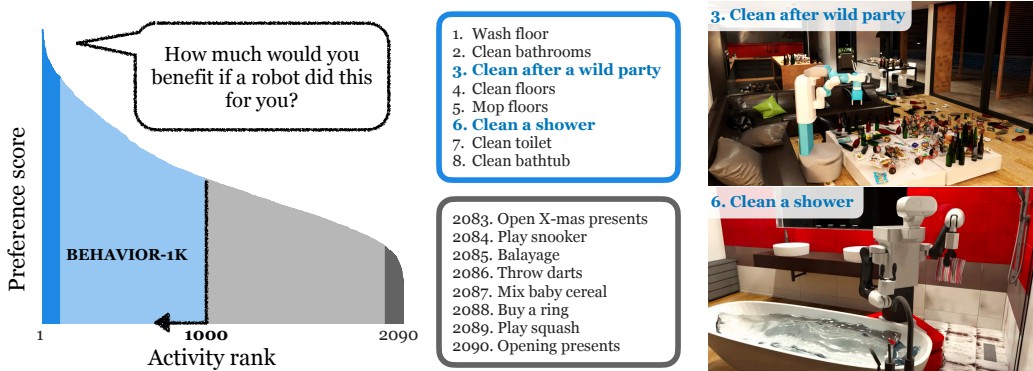

Figure 1: **Developing a Human-Centered Benchmark for Embodied AI.** Left: human preference score over 2,090 activities, ranked based on a survey on 1,461 participants. The distribution indicates the high **diversity** of needs and preferences of humans that should be reflected in a comprehensive benchmark. Middle: Example activities. Laborious activities are ranked the highest, while pleasurable ones are ranked the lowest. Right: visualization of two of the top 8 activities generated by our **realistic** OMNIGIBSON simulation environment.

range of daily activities based on participants' desire to delegate these activities to robots. We also ask layperson annotators to provide definitions of those activities. The survey reveals systematicity in what activities people want robots to do, but more importantly, highlights two key factors that we should prioritize when designing robotic benchmarks: **diversity** in the type of scenes, objects, and activities, and **realism** of the underlying simulation environments.

The most needed activities indicated by the survey range from 'wash floor' to 'clean bathtub.' Clearly, the diversity of these activities is far beyond what real-world robotics challenges may offer [35–42]. Developing simulation environments is a natural alternative: one can train and test robotic agents in many activities with diverse scenes, objects, and conditions efficiently and safely. However, for this paradigm to work, the activities have to be simulated realistically, reproducing accurately the circumstances that a robot may encounter in the real world. While significant progress in realism has been made in specific domains [43–45], achieving realism for a diverse set of activities remains a tremendous challenge, due to the effort required to provide realistic models and simulation features.

In this work, we present **BEHAVIOR-1K**, a Benchmark of 1,000 Everyday Household Activities in Virtual, Interactive, and Ecological Environments – the next generation of BEHAVIOR-100 [27]. BEHAVIOR-1K includes two novel components to address the demands for diversity and realism: the diverse **BEHAVIOR-1K DATASET** and the realistic **OMNIGIBSON** simulation environment. The BEHAVIOR-1K DATASET is a large-scale dataset comprising 1) a common-sense knowledge base for 1,000 activities with definitions in predicate logic (initial and goal conditions), as well as the objects involved, their properties, and their state transitions, and 2) high-quality 3D assets including 50 scenes and 5,000+ object models with rich physical and semantic annotations.

All activities in the BEHAVIOR-1K DATASET are instantiated in a novel simulation environment, OMNIGIBSON, which we build on top of Nvidia's Omniverse and PhysX 5 [46] to provide realistic physics simulation and rendering of rigid bodies, deformable bodies, and fluids. OMNIGIBSON expands beyond Omniverse's capabilities with a set of extended object states like temperature, toggled, soaked, and dirtiness. It also includes capabilities to generate valid initial activity configurations and discriminate valid goal solutions based on activity definitions. With all these realistic simulation features, OMNIGIBSON supports the 1,000 diverse activities in the BEHAVIOR-1K DATASET.

We evaluate state-of-the-art reinforcement learning algorithms [47, 48] in several activities of BEHAVIOR-1K, both with visuomotor control in the original action space, and with action primitives that leverage sampling-based motion planning [49]. Our analysis indicates that even a single activity in BEHAVIOR-1K is extremely challenging for current AI algorithms, and the baselines can only solve it with a significant injection of domain knowledge. Concretely, the difficulties derive in part from the length of BEHAVIOR-1K's activities and the complexity of the physical manipulation required. To calibrate the simulation-to-real gap of BEHAVIOR-1K, we provide an initial study on transferring solutions learned with a mobile manipulator in a simulated apartment to its real-world

| | BEHAVIOR-1K | BEHAVIOR-100 | AI2THOR Vis. Room Rearr. | TDW Transport | Rearr. T5 (Habitat) | ManipulaTHOR ArmPointNav | Interactive Gibson Benchmark | VirtualHome | ALFRED | Habitat 2.0 HAB | SAPIEN ManiSkill | Watch-And-Help | RFUniverse | Rearr. T2 (OCRTOC) | IKEA Furniture Assembly | RLBench | Metaworld | Robosuite | SoftGym | DeepMind Control State | OpenAIGym | Habitat 1.0 | Gibson |
|---|---|---|---|---|---|---|---|---|---|---|---|---|---|---|---|---|---|---|---|---|---|---|---|
| | | | | | | | Mobile manipulation | | | | | | | | | | Static manipulation | | | | | | Navigation | |
| Activities from human preference survey | ✔ | ✗ | ✗ | ✗ | ✗ | ✗ | ✗ | ✗ | ✗ | ✗ | ✗ | ✗ | ✗ | ✗ | ✗ | ✗ | ✗ | ✗ | ✗ | ✗ | ✗ | ✗ | ✗ |
| **Diversity** — Activities | **1000** | 100 | 1 | 1 | 1 | 1 | 2 | 549 | 7 | 3 | 4 | 5 | 5 | 5 | 1 | 50 | 1 | 5 | 10 | 28 | 8 | 2 | 3 |
| Scene types | **8** | 1 | 1 | 1 | 1 | 1 | 1 | 1 | 1 | 1 | 1 | 1 | Gibson | 1 | 1 | 1 | 1 | 1 | 1 | 1 | 1 | 1 | 1 |
| Scenes / rooms | **50/306** | 15/100 | -/120 | 15/105 | 55 static/- | -/30 | 10/- | 6/24 | -/120 | 1/6 | 1/- | 7/29 | Gibson | 1/- | 1/- | 1/- | 1/- | 1/- | 1/- | 1/- | 1/- | HM3D | 572 static |
| Object categories | **1265** | 391 | 118 | 50 | YCB | 150 | 5 | 308 | 84 | 41+YCB | 4 | 117 | UNK | 12+YCB | 73+ | 28 | 7 | 10 | 4 | 4 | 4 | Mpt. | N/A |
| Object models | **5215** | 1217 | 118 | 112 | YCB | 150 | 152 | UNK | 84 | 92+YCB | 162 | UNK | UNK | 101+YCB | 73+ | 28 | 80 | 10 | 4 | 4 | 4 | N/A | N/A |
| Objs. per activity | **3-47** | 3-34 | 5 | 7-9 | 2-5 | 2-3 | 10 | 1-24 | 2 | 5 | 1 | 2-8 | 1-6 | 5-10 | 1-2 | 1-2 | 1 | 1-3 | 1-3 | 1-3 | 1-3 | 0-1 | N/A |
| Diff. state changes required per activity | **2-11** | 2-8 | 4 | 4 | 4 | 2 | 1-3 | 1-7 | 2-3 | 1-2 | 1 | 1-3 | 1 | 1 | 1-3 | 1-4 | 4 | 1 | 1-3 | 1-2 | 1-2 | 1 | 1 |
| Infinite scene-agnostic instantiation | ✔ | ✔ | ✗ | ✗ | ✗ | ✗ | ✗ | ✗ | ✔ | ✔ | ✗ | ✗ | ✗ | ✗ | ✗ | ✗ | ✗ | ✗ | ✗ | ✗ | ✗ | N/A | N/A |
| **Realism** — Visual quality score | **3.20** | 1.69 | 1.73 | 1.65 | - | 1.73 | - | - | 1.73 | 1.74 | - | - | - | - | - | - | - | - | - | - | - | - | - |
| Kinematics, dynamics | ✔ | ✔ | ✔ | ✔ | ✔ | ✔ | ✔ | ✔ | ✗ | ✔ | ✔ | ✗ | ✔ | ✔ | ✔ | ✔ | ✔ | ✔ | ✔ | ✔ | ✔ | ✔ | ✔ |
| Continuous extended states (temp., wetness) | ✔ | ✔ | ✗ | ✗ | ✗ | ✗ | ✗ | ✗ | ✗ | ✗ | ✗ | ✗ | ✔ | ✗ | ✗ | ✗ | ✗ | ✗ | ✔ | ✗ | ✗ | ✗ | ✗ |
| Flexible materials | ✔ | ✗ | ✗ | ✗ | ✗ | ✗ | ✗ | ✗ | ✗ | ✗ | ✗ | ✗ | ✔ | ✗ | ✗ | ✗ | ✗ | ✗ | ✔ | ✗ | ✗ | ✗ | ✗ |
| Deformable bodies | ✔ | ✗ | ✗ | ✗ | ✗ | ✗ | ✗ | ✗ | ✗ | ✗ | ✗ | ✗ | ✔ | ✗ | ✗ | ✗ | ✗ | ✗ | ✔ | ✗ | ✗ | ✗ | ✗ |
| Realistic Fluid | ✔ | ✗ | ✗ | ✗ | ✗ | ✗ | ✗ | ✗ | ✗ | ✗ | ✗ | ✗ | ✔ | ✗ | ✗ | ✗ | ✗ | ✗ | ✔ | ✗ | ✗ | ✗ | ✗ |
| Thermal effects (steam/fire) | ✔ | ✗ | ✗ | ✗ | ✗ | ✗ | ✗ | ✗ | ✗ | ✗ | ✗ | ✗ | ✔ | ✗ | ✗ | ✗ | ✗ | ✗ | ✗ | ✗ | ✗ | ✗ | ✗ |
| Realistic action execution | ✔ | ✔ | ✔ | ✔ | ✔ | ✔ | ✔ | ✗ | ✗ | ✔ | ✔ | ✗ | ✔ | ✔ | ✔ | ✔ | ✔ | ✔ | ✔ | ✔ | ✔ | ✗ | ✗ |
| Benchmark focus: Task-Planning and/or Control | TP+C | TP+C | TP | TP+C | TP+C | TP+C | C | TP | TP | TP+C | C | TP | TP+C | TP+C | C | TP+C | C | C | C | C | C | C | C |

Table 1: **Comparison of Embodied AI Benchmarks:** BEHAVIOR-1K contains 1,000 diverse activities that are grounded by human needs. It achieves a new level of diversity in scenes, objects, and state changes involved. OMNIGIBSON provides realistic simulation of these 1,000 activities, including some of the most advanced simulation and rendering features such as fluid and deformable bodies. This table is extended from [27].

counterpart. We hope that the BEHAVIOR-1K benchmark, our survey, and our analysis will serve to support and guide the development of future embodied AI agents and robots.

## 2 Creating a Benchmark Grounded in Human Needs: A Survey Study

A significant amount of robotics research aspires to satisfy human needs, but those needs are typically assumed or speculated. Human-centric development requires direct information about what humans want from autonomous agents [31]. To create a benchmark that reflects these needs, we conduct a survey targeting the general U.S. population that asks: *what do you want robots to do for you?* The survey sources around 2,000 activities from time-use surveys [50–52], which record how people spend their time, and from WikiHow articles [53]. We conduct the survey on Amazon Mechanical Turk with a total of 1,461 respondents (demographics in Appendix A.3) and fifty 10-point Likert scale responses per activity.

Survey results are summarized in Fig. 1 (left), in which we rank the activities based on their human preference score. The full list of ranked activities can be found on our website. The distribution shows large statistical dispersion (Gini index=0.158): humans want robots to perform a wide range of activities, from cleaning chores to cooking large feasts. Tedious tasks like "scrubbing the bathroom floor" score the highest, while recreational activities like game-play score the lowest. There are around 200 cleaning activities and over 200 cooking activities, among many other categories.

BEHAVIOR-1K activities include the 909 activities with the highest human preference scores and 91 activities from BEHAVIOR-100 [27], altogether the top-ranked 1000 activities. BEHAVIOR-1K sets itself apart from other embodied AI benchmarks by sourcing from time-use surveys and using survey data to prioritize activities considered most important and useful by humans, and by including a tremendously diverse set of activities.

## 3 Related Work: Embodied AI Benchmarks

We provide an extensive comparison between BEHAVIOR-1K, and other embodied AI benchmarks in simulation [17–26] in Table 1. We include a number of factors that contribute to diversity and realism and observe a significant step forward with BEHAVIOR-1K. First, no other benchmark grounds their activity set on the needs of lay people. Other benchmarks often target a relatively restricted set of activities, and their simulators are realistic only in the relevant aspects for those tasks. In fact, we often observe a diversity-realism tradeoff. For instance, instruction-following benchmarks such as VirtualHome [20] and ALFRED [20, 21] are diverse in the number of scenes, objects, and state changes, but offer a limited low-level physical realism. On the other hand, household rearrangement benchmarks such as Habitat 2.0 HAB [26], TDW Transport [19], and SAPIEN ManiSkill [54, 55] support realistic action execution and accurate physics simulation for rigid bodies, but only include a handful of tasks. Similarly, SoftGym [45] and RFUniverse [56] have the closest simulation features and hence realism to OMNIGIBSON, but they also lack the task diversity needed to support the development of human-centered general robots.

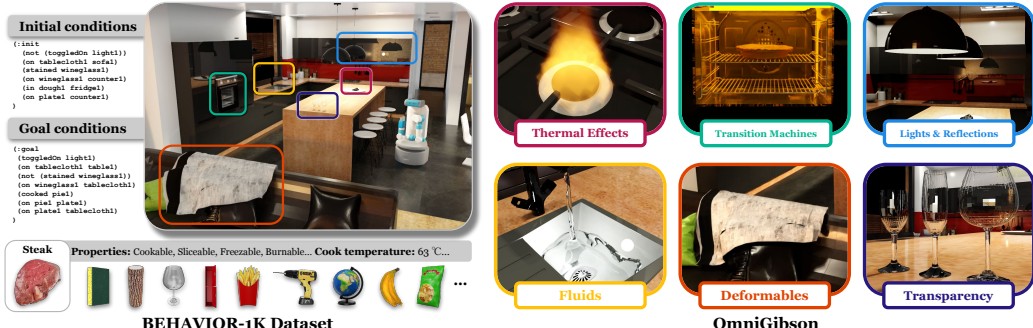

Figure 2: **Elements of BEHAVIOR-1K.** Our benchmark comprises two elements: BEHAVIOR-1K DATASET and OMNIGIBSON. Left: BEHAVIOR-1K DATASET includes 1,000 BDDL activity definitions (top left), 50 realistic and diverse scenes (top right), and 5,000+ objects with properties annotated in the knowledge base (bottom). Right: OMNIGIBSON provides the necessary functionalities to realistically simulate the 1000 activities, including thermal effects such as fire/steam/smoke (top left), fluid dynamics (bottom left), functional machines for transition rules (top center), deformable bodies/cloths (bottom center), realistic lighting and reflections (top right), and transparency rendering (bottom right). Together, they constitute a concrete, realistic instantiation of an everyday activity like CookingDinner in simulation.

The most similar benchmark to us is the previous generation BEHAVIOR-100 [27]. BEHAVIOR-100 brought forward several beneficial design choices that we inherit in BEHAVIOR-1K such as the activity sources (ATUS [50]), activity definition logic language, and evaluation metrics. However, it fell short in the diversity and realism necessary to support a human-centered embodied AI benchmark in simulation, dimensions where BEHAVIOR-1K achieves unmatched levels. While BEHAVIOR-100 comprises 100 activities selected by researchers, our BEHAVIOR-1K increases diversity by one order of magnitude, to 1,000 activities, that are grounded in human needs thanks to our unique survey. Furthermore, BEHAVIOR-100 includes only 15 scenes (all houses) and 300+ object categories, while BEHAVIOR-1K increases to 50 scenes (houses, stores, restaurants, offices, etc.) and 1,200+ object categories. In terms of realism, BEHAVIOR-1K extends the simulatable physical states and processes with OMNIGIBSON: fluids, flexible materials, mixing substances, etc. The realism achieved in rendering by OMNIGIBSON for BEHAVIOR-1K is also significantly higher than what was possible in BEHAVIOR-100 and other benchmarks (see Fig. 3).

## 4 BEHAVIOR-1K DATASET

Once activities have been sourced to reflect human needs, they need to be concretely defined and instantiated the way they would occur in the real world. We build the BEHAVIOR-1K DATASET, which includes a knowledge base of crowdsourced activity definitions with relevant objects and object states, and a large-scale repository of high-quality, interactive 3D models.

We crowdsource concrete definitions of activities in the form of BEHAVIOR Domain Definition Language (BDDL) [27]. BDDL is based on predicate logic and designed to be accessible for laypeople to describe concrete initial and goal conditions for a given activity. Unlike geometric, image/video, or experience goal specifications [17, 18], BDDL definitions are in terms of objects and object states, allowing annotators to define at an intuitive semantic level. The semantic symbols also capture the fact that multiple physical states might be valid initializations and solutions to an activity. See Listing 1, 2 and 3 in Appendix for example definitions.

The object and object state spaces that activity definitions are built upon are annotated to be ecologically plausible. The object spaces are derived from 5,000 WikiHow articles for the 1,000 activities and mapped to 1,484 WordNet [57] synsets. Through crowdworkers, students, and GPT-3 [58], we also associate each object with our fully simulatable object states: for example, apple is associated with cooked and sliced, but not toggledOn. Many object-property pairs are augmented with parameters, e.g. "cooked temperature for apples", taking advantage of OMNIGIBSON's continuous extended states to make activities especially realistic. Finally, annotators and researchers also create transition rules, e.g. turning tomatoes and salt into sauces, or requiring sandpaper to remove rust. The result is a knowledge base of tens of thousands of elements underlying 1000 ecologically plausible activity definitions. We ensure annotation quality by having five experienced machine learning annotators

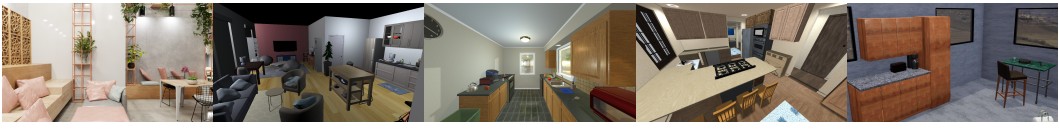

| **OMNIGIBSON** | Habitat 2.0 | AI2-THOR | iGibson 2.0 | ThreeDWorld |
|:---:|:---:|:---:|:---:|:---:|
| **3.20 ± 1.23** | 1.74 ± 1.33 | 1.73 ± 1.37 | 1.69 ± 1.24 | 1.65 ± 1.23 |

Figure 3: **Comparison of Visual Realism:** We evaluate OMNIGIBSON's visual realism against other simulation environments by running a survey with 60 human subjects. We ask them to rank the realism of sampled images from each environment with a score of 5 (most realistic) to 1 (least realistic). We report the mean and standard deviation and show a sampled image from the study. We observe that the participants consider OMNIGIBSON to be significantly more visually realistic than all other environments. See Appendix E.2 for more info.

verify a subset of all types of annotations and receive extremely high approval rates (>96.8%). See Appendix B for more details about the knowledge base.

The diversity of these activity definitions requires diverse object and scene models. On top of the 15 house scenes from BEHAVIOR-100 [27], we acquire 35 fully interactive scenes across diverse scene types, such as gardens, offices, restaurants, and stores, that are essential for everyday activities. This is unprecedented compared to other benchmarks (see Table 1). We also acquire 5,000+ object instances across 1,200+ categories required by the activities, and annotate rich physical (e.g., friction, mass, articulation) and semantic properties (e.g., category) for each object. Representative scene and object models can be seen in Fig. 2. More details of 3D models can be found in Appendix D.

## 5 OMNIGIBSON: Instantiating BEHAVIOR-1K with Realistic Simulation

BEHAVIOR-100 is implemented in iGibson 2.0 [59]; however, realistic simulation of the diverse activities in BEHAVIOR-1K is beyond the capability of iGibson 2.0. We present a novel simulation environment, OMNIGIBSON, that provides the necessary functionalities to support and instantiate BEHAVIOR-1K. OMNIGIBSON is built on top of Nvidia Omniverse and PhysX 5, providing the simulation of not only rigid bodies, but also deformable objects, fluids, and flexible materials (see Fig. 4), while generating highly realistic ray-traced or path-traced virtual images (see Fig. 3). These features significantly boost the realism of BEHAVIOR-1K compared to other benchmarks.

Similar to BEHAVIOR-100, OMNIGIBSON also simulates additional, non-kinematic extended object states (e.g. temperature, soaked level) based on heuristics (e.g. temperature increases when being next to a heat source that is toggled on). OMNIGIBSON also implements the functionalities to generate infinite valid physical configurations that satisfy the activities' initial conditions as logical predicates (e.g. food is frozen), and to evaluate their goal conditions (e.g. food is cooked and onTop of a plate, the cloth is folded) based on the object's physical states (pose and joint configuration) and extended states. OMNIGIBSON natively supports randomization during scene initialization, and can sample amongst object models and their poses/states. The full details of extended object states and logical predicates that OMNIGIBSON supports can be found in Appendix E.1.

Many everyday tasks are difficult to simulate because they require modeling complex physical processes, such as folding a towel or pouring a glass of water. OMNIGIBSON unlocks them by supporting realistic simulation of fluids, deformable bodies, and cloths (see Fig. 2). Indeed, without these features, over half of BEHAVIOR-1K activities would not be simulatable, highlighting how crucial these features are for capturing everyday activities. OMNIGIBSON also captures multiple physical processes that are not natively simulatable by Omniverse, such as baking pies or pureeing vegetables. Aside from the extended states mentioned above, we also design a modular *Transition Machine*, which specifies custom transitions between groups of objects when specified conditions are met. For example, a dough placed inside an oven that reaches a certain temperature threshold will turn into a pie. This further expands OMNIGIBSON's capacity to simulate complex, realistic activities that would otherwise be intractable to fully simulate physically.

## 6 Experiments: Evaluating Embodied AI Solutions in BEHAVIOR-1K

In our experiments, we aim to answer three questions: How do existing vision-based robot learning algorithms perform in BEHAVIOR-1K, and what assumptions have to be made to improve their success? What elements of the activities are the most problematic for current AI? What are the main

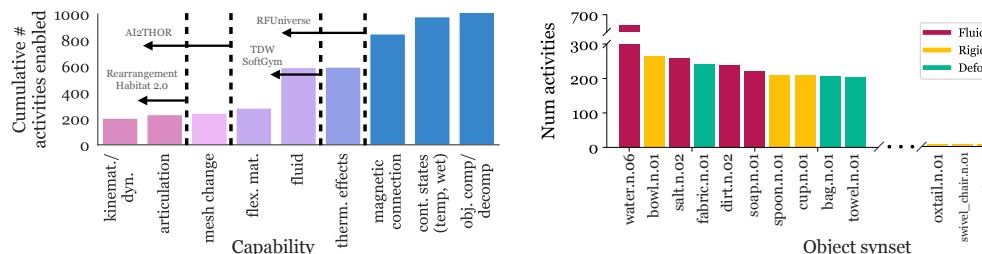

Figure 4: **Objects and States in Activity Definitions:** Left: the number of activities unlocked by each simulation capability that OMNIGIBSON has. None of the other simulation environments are sufficient to fully support BEHAVIOR-1K, e.g. Habitat 2.0 can support only 23% of the activities. Right: the number of activities that require each object synset (category). Several top-10 object synsets are fluids and flexible materials, necessitating the development of OMNIGIBSON. As we expect, the object synsets also follow a long-tail distribution: most objects are involved in only a few activities.

| Method | Policy Features | | Task success rate | | |
|---|---|---|---|---|---|
| | Primitives | History | StoreDecoration | CollectTrash | CleanTable |
| RL-VMC | ✗ | ✗ | $0.0 \pm 0.0$ | $0.0 \pm 0.0$ | $0.0 \pm 0.0$ |
| RL-Prim. | ✓ | ✗ | $0.48 \pm 0.06$ | $0.42 \pm 0.02$ | $0.77 \pm 0.08$ |
| RL-Prim.Hist. | ✓ | ✓ | $0.55 \pm 0.05$ | $0.63 \pm 0.03$ | $0.88 \pm 0.02$ |

Table 2: Task success rates across three baseline methods. RL-VMC with end-to-end visuomotor control completely fails to solve any of the activities, whereas RL-Prim. and RL-Prim.Hist. with action primitives are able achieve decent performance. Memory of observations helps in longer horizon activities (e.g. CollectTrash).

sources of the sim-real gap in BEHAVIOR-1K/OMNIGIBSON? Our goal is to indicate promising research directions to improve AI's performance in BEHAVIOR-1K activities in simulation and, ultimately, in the real world.

### 6.1 Evaluating BEHAVIOR-1K Solutions in OMNIGIBSON

**Experimental Setups.** We selected three paradigmatic activities for our experiments: CollectTrash, where the agent gathers empty bottles and cups, and throws them into a trash bin (rigid body manipulation); StoreDecoration, where the agent stores items into a drawer (articulated object manipulation); and CleanTable, where the agent wipes a dirty table with a soaked piece of cloth (manipulation of flexible materials and fluids). We evaluate three different baselines based on state-of-the-art reinforcement learning algorithms (RL) [60]:

- RL-VMC, a visuomotor control (from image to low-level joint commands) RL solution based on Soft Actor-Critic (SAC) [48];

- RL-Prim., a RL solution based on PPO [47] that leverages a set of action primitives based on a sampling-based motion planner [61, 62, 49] (pick, place, push, navigate, dip and wipe). The policy outputs a discrete selection of a primitive applied on an object;

- RL-Prim.Hist., a variant of RL-Prim. that takes in the history observations (3 steps) as additional inputs to help disentangle similar-looking states.

All agents are trained with a sparse task success reward without any reward engineering. Following the metrics proposed in BEHAVIOR-100 [27], we report the success rate and efficiency metrics (distance traveled, time invested, and disarrangement caused) in Table 2 and 3, and the success score Q in Table A.13 in Appendix.

Grasping is a challenging research topic on its own. To facilitate our experiments, we adopt an assistive pick primitive that creates a rigid connection between the object and the gripper if grasping is requested when all fingers are in contact with the object, a stricter form of *StickyMitten* used in prior works [26, 63, 64]. Furthermore, to accelerate training, the action primitives check only the feasibility (e.g., reachability, collisions) of the final configuration, e.g. the grasping pose for pick or the desired location for navigate. If kinematically feasible, the action primitives will directly set the robot state to the final configuration, and continue to simulate from then on. We include an ablation

| Method | Metrics in CollectTrash | | |
|--------|---------------|--------------|--------------|
| | Dist. Nav. [m] | Sim. Time [s] | Kin. Dis. [m] |
| RL-VMC | $27.58 \pm 5.95$ | $16.67 \pm 0.00$ | $0.00 \pm 0.00$ |
| RL-Prim. | $17.98 \pm 2.35$ | $13.95 \pm 5.14$ | $12.34 \pm 5.01$ |
| RL-Prim.Hist. | $15.33 \pm 2.70$ | $12.48 \pm 3.68$ | $10.82 \pm 3.90$ |

Table 3: Efficiency metrics across three base-line methods. RL-VMC has low spatial and temporal efficiency because it fails to learn, whereas history information helps remove redundant actions and improve efficiency.

| Phys. Realism | | Task success rate | | |
|---------------|-------------|----------------|--------------|-----------|
| Grasping | Full Motion | StoreDecoration | CollectTrash | CleanTable |
| ✔ | ✔ | $0.0 \pm 0.0$ | $0.0 \pm 0.0$ | $0.0 \pm 0.0$ |
| ✘ | ✔ | $0.46 \pm 0.04$ | $0.36 \pm 0.08$ | $0.73 \pm 0.03$ |
| ✘ | ✘ | $0.48 \pm 0.06$ | $0.42 \pm 0.02$ | $0.77 \pm 0.08$ |

Table 4: Ablation study of RL-Prim. on the impact of removing the simplifying assumptions of grasping and motion execution during evaluation. We observe a large drop in performance when enabling fully physics-based grasping, but not when enabling full trajectory motion execution.

analysis of the effect of these assumptions and simplifications in our evaluation (see Table 4). Further details about our training and evaluation setup can be found in Appendix F.

**Results: Task Completion.** Table 2 contains task success rates across our baseline methods. The extreme long-horizon in our activities causes the visuor-motor control (RL-VMC) policy to fail in all three activities, potentially due to problems such as credit assignment [65], deep exploration [66, 67], and vanishing gradients [68] as reported by prior works. Our baselines with time-extended action primitives (RL-Prim. and RL-Prim.Hist.) obtain better success, achieving over 40% success rates across all three activities. We observe that longer-horizon activities are more challenging: while CleanTable can be accomplished by executing the optimal sequence of 6 primitive steps, CollectTrash requires at least 16. This supports the idea that some form of action-space abstraction must be necessary to solve long-horizon activities of BEHAVIOR-1K, as others reported [27, 26, 69]. When analyzing the role of memory, we observe a sizable performance gain from RL-Prim. to RL-Prim.Hist., especially in long-horizon activities with aliased observations such as CollectTrash. In this task, when the robot is looking at the trash bin, it needs additional information to know what location has been cleaned already in order to proceed to other locations. Our results indicate that memory will play a critical role for embodied AI in long-horizon BEHAVIOR-1K activities.

**Results: Efficiency.** In addition to success, efficiency is also critical in the evaluation of embodied AI: a successful policy in simulation may be infeasible in the real world if it takes a long time or wastes too much energy. In Table 3, we report the results with three efficiency metrics proposed by Srivastava et al. [27]. We observe that the use of memory (RL-Prim.Hist.) improves efficiency across all metrics: distance navigated (Dist. Nav.), simulated time (Sim. Time), and kinematic object disarrangement (Kin. Dis.), i.e., amount of object displacement due to robot motion.

We also evaluate to what extent the simplifications we introduce in physics and actuation (grasping, motion execution) during training impact the performance of RL-Prim. during evaluation when these simplifications are removed. We report the results in Table 4. We observe a radical performance drop after enabling fully physics-based grasping during evaluation. Grasping is thus a critical component of any embodied AI task and researchers should be careful when simplifying its execution during training. While OMNIGIBSON supports fully physics-based grasping, designing a pick action primitive for arbitrary objects that leverages fully physics-based grasping is by itself an open research problem that we leave for future work. In contrast, there is much less performance drop after enabling full trajectory motion execution during evaluation. This result supports our hypothesis that it is reasonable to accelerate the training process by assuming that motion planning is likely to provide viable paths in free space during evaluation.

### 6.2 Evaluating BEHAVIOR-1K Solutions on a Real Robot

We performed a series of experiments with a real robot to answer the question: *what are the main sources of discrepancy between our realistic simulation and the real world?* To that end, we used a real-world counterpart of the simulated scene of a mockup apartment for the CollectTrash activity. We scanned the apartment and converted it into a virtual, interactive scene. We use a real bi-manual mobile manipulator Tiago, and leverage the RGB-D images from its onboard sensors and a YOLOv3 object detector [70, 71] to localize the objects in 3D space for manipulation. For navigation, the robot localizes with a particle filter [72] based on two LiDAR sensors and a map of the apartment. The action primitives are implemented with the same sampling-based motion planning algorithm as in simulation [62, 49] with additional tuning. We evaluate two strategies for selecting action primitives in the real world: an optimal policy based on human input, and a vision-based policy (RL-Prim.) trained in OMNIGIBSON. To facilitate sim-to-real transfer, during training we additionally applied image-based data augmentation to the observations based on a prior work [73] (see Appendix G for

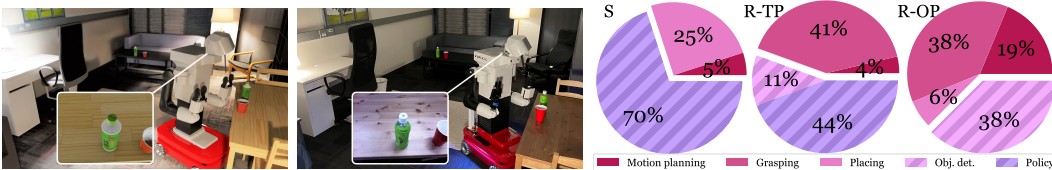

Figure 5: **Characterizing the Sim-Real Gap:** Left: a side-by-side comparison of the simulated and the real scene, including virtual and real images obtained by the robot. While the high resolution images are extremely similar, the mismatch in wooden texture and camera properties causes a sizable gap in the visual input to the agent. Right: source of failure in Simulation (S, left) and in Real-world with a Trained Policy in OmniGibson (R-TP, middle) and an Optimal Policy (R-OP, right) due to actuation (solid color) or perception (striped). In simulation, without full simulation of grasping (see Sec. 6.1), policy failures (i.e., selecting the wrong action primitive) dominate. On the real robot, grasping is one of the main causes of error, as well as perception issues (policy errors with the trained visual policy, object detection errors with the optimal policy).

further details). With the optimal policy, we evaluate the gap in actuation between the simulated and the real robot; with the learned policy, we also evaluate the gap in visual perception. We achieve different success rates in simulation (50 runs, ~40% success) and in real world with optimal (27 runs, ~22%) and trained policies (26 runs, 0%), hinting on a sim-real gap that we analyze below.

The failure cases are depicted in Fig. 5 (right). We observe that the majority of failures in simulation are due to the visual policy (perception), while others are caused by stochasticity in the `place` primitive and the sampling-based motion planner. The reason why none of the failures are due to grasping is because in simulation we evaluate with the assistive `pick` primitive. Grasping is fairly difficult in the real world, contributing to around 40% of the failures for both the trained and the optimal policy. For the learned policy, 44% of the errors come from the visual policy selecting the wrong action primitive due to the differences between the simulated and the real images. The visual discrepancy results from unmodeled effects such as the real camera's poor dynamic range (see Fig. 5, left and middle) and imperfect object modeling (e.g. the exact wooden texture and the surface reflectivity of the tables), which can be alleviated by more targeted domain randomization. Interestingly, several manipulation failures on the real robot are caused by unfavorable robot base placement resulting from navigation inaccuracies in the previous timestep. This compounding source of error is not present in simulation because we assume perfect localization and execution. We believe this analysis provides relevant information about the main sources and severity of the sim-real gap in BEHAVIOR-1K in OmniGibson, and provide insights for future research avenues. Our plan is to use some of these insights to create novel sim-real solutions that make progress on BEHAVIOR-1K.

## 7 Discussion and Limitations

We presented BEHAVIOR-1K, a benchmark for embodied AI and robotics research with realistic simulation of 1000 diverse activities grounded in human needs. BEHAVIOR-1K comprises two elements: BEHAVIOR-1K DATASET, a semantic knowledge base of everyday activities, and a large-scale 3D model library; OmniGibson, a simulation environment that provides realistic rendering and physics for rigid/deformable objects, flexible materials and fluids. In our evaluation, we observed that BEHAVIOR-1K is an extremely challenging benchmark: solving these 1,000 activities autonomously is beyond the capability of current state-of-the-art AI algorithms. We studied and attempted to solve a handful of the activities with action primitives in order to gain insights into the most challenging components, providing a starting point for other researchers to work on our benchmark. Similarly, we explored the sources of the sim-real gap by creating a digital twin of a real-world mock apartment, and by performing rigorous evaluation and analysis of policies in both simulation and the real world with a simulated and real mobile manipulator.

**Limitations:** We inherit several limitations from our underlying physics and rendering engine, Nvidia's Omniverse. In OmniGibson, we trade off rendering speed for visual realism (ray-traced), reaching around 60 fps for a house scene of around 60 objects (v.s. around 100 fps in iGibson 2.0 [59]). We are actively working on performance optimization. Another limitation is that we only include activities that do not require interactions with humans. Realistic simulation of humans (behavior, motion, appearance) is extremely challenging and an open research area. We plan to include simulated humans when the technology becomes more mature. Finally, there is still room for improvement in OmniGibson to further facilitate sim2real transfer, such as incorporating noise models of perception and actuation.

**Acknowledgments**

This work was done in part when Chengshu Li, Josiah Wong, and Michael Lingelbach were interns at Nvidia Research. The work is in part supported by the Stanford Institute for Human-Centered AI (HAI), the Toyota Research Institute (TRI), NSF CCRI #2120095, NSF RI #2211258, NSF NRI #2024247, ONR MURI N00014-22-1-2740, ONR MURI N00014-21-1-2801, Amazon, Bosch, Salesforce, and Samsung. Ruohan Zhang is partially supported by Wu Tsai Human Performance Alliance Fellowship. Sanjana Srivastava is partially supported National Science Foundation Graduate Research Fellowship Program (NSF GRFP).

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
