# OpenReview forum: "BEHAVIOR-1K: A Benchmark for Embodied AI with 1,000 Everyday Activities and Realistic Simulation"
_robot-learning.org/CoRL/2022/Conference — CoRL 2022 Oral_

### Official Review · Reviewer_HvRe · 2022-07-27

**Originality:** Good
**Technical Quality:** Excellent
**Clarity Of Presentation:** Excellent
**Impact:** 4

**Recommendation:**

Strong Accept: I recommend accepting the paper and will argue for my recommendation even if other reviewers hold a different opinion.

**Summary:**

The authors present a robot simulation benchmark that is comprised of 1000 activities grounded in 50 scenes. The benchmark comes with a physics simulator that allows for simulating the activities in a realistic environment. The activities are picked based on a survey of everyday users, making the benchmark human-centric.

**Issues:**

- Can a scene be randomized? For instance for the "cleaning after a wild party task", does the scene always look the same? Or are for instance the objects randomly placed on the table? The latter point would be of course important for the realism. I think the authors should clarify this in the paper.

**Quality Of The Limitations Section:**

Limitations are addressed clearly

**Reviewer Expertise:**

4: The reviewer is confident but not absolutely certain that the evaluation is correct

**Robotics Focus:**

Sufficient demonstration on hardware

**Strengths And Weaknesses:**

Strengths
- Compared to existing benchmarks, behavior-1k seems to be a big leap forward given its sheer size in terms of number of activities, objects, scenes and annotations.
- Very detailed comparison with existing benchmarks

Weaknesses
- For the sim2real transfer, it is difficult to assess
how physical object interactions skills can be transferred to the real world (e.g., complex force interactions beyond grasping). However, I don't really see this as a weakness of the presented benchmark but a weakness of simulators in general.
- Difficulty of grasping has been "bypassed" in the experiments

**Summary Of Recommendation:**

I think the presented benchmark is very valuable and of high interest for the robotics community. The paper is written well and the comparison with existing benchmarks is very detailed. The experiments show that an agent trained in the simulated environment can in principle perform a task also in the real world. The authors also did a good job in discussing the existing limits in such a sim2real scenario.

---

> ### Author Response · Authors · 2022-08-25
> **Response to Reviewer HvRe**
>
> **Comment:**
>
> Thank you for your constructive comments and helpful feedback.
>
> We appreciate your recognition of the diversity and scale of our dataset and annotations, our detailed and honest comparison with existing benchmarks, and our attempt to deploy a trained policy on a real robotic platform.
>
> **Q1: How physical object interactions skills can be transferred to the real world? The difficulty of grasping has been "bypassed" in the experiments.**
>
> A1: We would like to clarify that our physics simulation is realistic for robot-object interactions. Our simulator supports two modes of grasping: physical grasping and simplified grasping. Physical grasping is fully simulated by the underlying PhysX 5 physics engine and strictly relies on friction between the gripper and the object for grasping, as in the real world. Simplified grasping artificially creates a rigid connection (“sticky-mitten”) between the gripper and the object when certain conditions are met, and is therefore less realistic.
>
> In our experimental evaluation, we adopted a pick primitive with simplified grasping to facilitate policy learning, given the challenging nature of our long-horizon household tasks. This is why the grasping shown in the supplemental video might not seem realistic. In the attached zip file, we included two new videos (“**physical_grasping_rigid.mp4**” and “**physical_grasping_cloth.mp4**”) that showcase (non-simplified) physical grasping of a rigid object (bottle) and a soft cloth object (towel), where the grasping behavior is a lot more realistic.
>
> We believe the high physical realism for grasping and low-level manipulation of OmniGibson will provide a useful tool for the robotics community that targets real-world applications to develop and evaluate new solutions.
>
> **Q2: Can a scene be randomized?**
>
> A2: Yes. One of the main highlights of our benchmark is the ability to sample infinite variations of the scene for a given task. For example, for “cleaning after a wild party task”, let’s say the initial condition includes “OnTopOf(beer_bottle, kichen_counter)”. Our benchmark is able to programmatically sample 1) which beer bottle 3D model to use (we have many!), 2) which kitchen counter to place it on (there might be multiple in the kitchen), 3) the exact location to place it, and 4) the orientation of the bottle. We fully agree that the ability to randomize the scene is essential for improving the realism of the environment and for training robust policies, and we have clarified this in the revised paper.
>
>
> **Zip File:**
>
> /attachment/d86185bc176f3216faf2430ac38b5b5adce713fd.zip

---

> > ### Author Response · Authors · 2022-08-27
> > **Looking forward to the discussion**
> >
> > Dear Reviewer HvRe,
> >
> > Thank you again for your constructive reviews, which have helped us improve the quality and clarity of the paper. We hope that we have been able to address your concerns and questions regarding the realism of grasping, and how users can randomize a scene. As we approach the end of the rebuttal discussion period, please don't hesitate to let us know if you have any additional questions or comments. We look forward to the discussion!
> >
> > Thank you for your time.
> >
> > Warmest,
> >
> > Authors

---

### Official Review · Reviewer_NBGD · 2022-07-29

**Originality:** Excellent
**Technical Quality:** Excellent
**Clarity Of Presentation:** Excellent
**Impact:** 4

**Recommendation:**

Strong Accept: I recommend accepting the paper and will argue for my recommendation even if other reviewers hold a different opinion.

**Summary:**

This paper is a combination of two contributions. First, a crow-sourced database of activities based on the needs of robots in human-centered activities. This large dataset involves the predicates of the task, the objects involved in the task, and their high-fidelity models.
Second, a realistic simulator that is built on top of Nvidia Omniverse and PhysX 5, which supports non-rigid bodies and ray-tracing.

The paper is well-motivated and clearly presented. The large-scale survey on robot activities was not only done thoughtfully but also goes well beyond similar previous attempts in the literature. The dataset, simulator, and insights are valuable tools and knowledge worth sharing with the robotics community, in particular, with those working with service robots.

**Issues:**

In the experimental section, the simulation seems to skip the fine interactions of grasping. Does it assume that real-world grasping of an object is going to work out of the box? In that sense, is the simulator useful to learn low-level fine-grained grasping policies or is it aimed at task-level (high-level) discrete actions?

Minor comments
================
1) It is nice to see that the simulator can address fluids, deformable, and transparent objects, although such capabilities have not been used in the paper.
Related to the fine-grained policies, what are the prospects of using OmniGibson to train policies based on such non-rigid elements? Does the simulator provide fidelity enough? And how easy it is for the end-user to tune the physics of the simulator?
2) Among the 1000 activities, is there any whose simulation requires the use of the non-rigid body capabilities of OmniGibson? And if so, how easy is it to set up a simulation scenario for such tasks (e.g. baking a pie?)

**Quality Of The Limitations Section:**

Limitations are addressed clearly

**Reviewer Expertise:**

3: The reviewer is fairly confident that the evaluation is correct

**Robotics Focus:**

Sufficient demonstration on hardware

**Strengths And Weaknesses:**

The work is quite refined and involved a large-scale effort that is difficult to carry out by small laboratories. Besides the dataset and the simulator, the paper also deployed a trained policy in a real environment as part of the experimental section, although it felt a bit rushed. Such a sim2real investigation deserves a dedicated paper, but nevertheless, it is a notable attempt of the paper to provide such experiments; which provides evidence of its usefulness for the practicioner.

**Summary Of Recommendation:**

Due to the large-scale work that is difficult to carry out by small laboratories, and given the unique database which includes the insights provided by a crowd-sourced effort, this paper represents a valuable contribution to the community.

---

> ### Author Response · Authors · 2022-08-25
> **Response to Reviewer NBGD**
>
> **Comment:**
>
> Thank you for your constructive comments and helpful feedback.
> We appreciate your acknowledgment that our dataset, simulator, and insights from experimental evaluation provide value to the community and your recognition of our attempt to deploy a trained policy on a real robotic platform.
>
> **Q1: The simulation seems to skip the fine interactions of grasping. Is the simulator useful for learning low-level fined-grained manipulation or high-level task planning?**
>
> A1: We believe the answer is both.
>
> We would like to clarify that our physics simulation is realistic for robot-object interactions. Our simulator supports two modes of grasping: physical grasping and simplified grasping. Physical grasping is fully simulated by the underlying PhysX 5 physics engine and strictly relies on friction between the gripper and the object for grasping, as in the real world. Simplified grasping artificially creates a rigid connection (“sticky-mitten”) between the gripper and the object when certain conditions are met, and is therefore less realistic.
>
> In our experimental evaluation, we adopted a pick primitive with simplified grasping to facilitate policy learning, given the challenging nature of our long-horizon household tasks. This is why the grasping shown in the supplemental video might not seem realistic. In the attached zip file, we included two new videos (“**physical_grasping_rigid.mp4**” and “**physical_grasping_cloth.mp4**”) that showcase (non-simplified) physical grasping of a rigid object (bottle) and a soft cloth object (towel), where the grasping behavior is a lot more realistic.
>
> In OmniGibson, our users have the flexibility to configure the desired action abstraction level (high-level, discrete action primitives v.s. low-level joint commands) and grasping mode (physical v.s. simplified) based on their own use cases. Hence, we believe the simulator is useful for learning both low-level fine-grained manipulation and high-level task planning.
>
> **Q2: Prospects of using OmniGibson to train policies based on non-rigid elements such as fluids, deformables and transparent objects.**
>
> A2: We’d like to highlight that the Clean Table task in our experimental evaluation includes fluids and deformables: the robot needs to grasp a soft towel and dip it into the water in the sink. The trained policy is able to achieve around an 88% success rate, which is promising. The simulation fidelity of these non-rigid elements appears very decent visually, as shown in our original supplementary video. Omniverse is the best simulation platform option by far that we can find in the field for non-rigid elements. In the future, we are hoping to conduct sim-to-real policy transfers that involve these non-rigid elements.
>
> **Q3: Among the 1000 activities, is there any whose simulation requires the use of the non-rigid body capabilities of OmniGibson? And if so, how easy is it to set up a simulation scenario for such tasks (e.g. baking a pie?)**
>
> A3: We’d like to highlight in Fig. 4 that more than half of the 1000 activities require non-rigid body capabilities of OmniGibson (e.g. fluids, deformables, mesh changes, object composition/decomposition), which is the main motivation for developing OmniGibson. If you would like to set up one of our 1000 activities, it’s as easy as a one-line change of the activity name in the config file. If you would like to create a new task, you can specify the initial conditions and goal conditions in a new BDDL file, and run our automatic sampling script to generate a valid scene initialization.
>
> **Q4: How easy is it for the end-user to tune the physics of the simulator?**
>
> A4: Our code provides easy-to-use APIs for users to tune the parameters of the physics of the simulator, such as the physics timestep, gravity, solver type and iterations, friction type and offset, and whether to enable continuous collision detection.
>
>
> **Zip File:**
>
> /attachment/6f40b2b483152e142899ba018838d5492464e0c6.zip

---

> > ### Author Response · Authors · 2022-08-27
> > **Looking forward to the discussion**
> >
> > Dear Reviewer NBGD,
> >
> > Thank you again for your constructive reviews, which have helped us improve the quality and clarity of the paper. We hope that we have been able to address your concerns and questions regarding the realism in fine-grained manipulation, the research purpose of our simulator, the motivation for developing and the prospect of using non-rigid elements, and how users can set up tasks and tune the physics. As we approach the end of the rebuttal discussion period, please don't hesitate to let us know if you have any additional questions or comments. We look forward to the discussion!
> >
> > Thank you for your time.
> >
> > Warmest,
> >
> > Authors

---

### Official Review · Reviewer_Qfw6 · 2022-07-29

**Originality:** Very Good
**Technical Quality:** Fair
**Clarity Of Presentation:** Excellent
**Impact:** 3

**Recommendation:**

Weak Accept: I recommend accepting the paper, but will not argue for my recommendation if the majority of other reviewers have a different opinion.

**Summary:**

BEHAVIOR-1k has a 1k dataset with everyday activities selected based on the survey. This dataset is made from OMNIGIBSON, a realistic physics simulator providing comprehensive types of objects. To provide the usefulness of this dataset, the author trained 3 types of RL-algorithms in their simulation environment and performed a sim2real using a baseline RL algorithm.


**Issues:**

Robot-object interactions do not seem realistic. Can the author explain the reason and the way to improve this?
- At 6’30’’, the gripper is not holding the pumpkin properly ( not a force closure), but the simulator considers that the robot is holding the pumpkin tight. I do not think we can train any pick-and-place algorithms works in real life based on this simulator because it ignores many fundamental physics for manipulation.
- Slowing down the play speed (x0.5) at 7’39’’, the robot gripper penetrates the towel.



**Quality Of The Limitations Section:**

Limitations are addressed clearly

**Reviewer Expertise:**

3: The reviewer is fairly confident that the evaluation is correct

**Robotics Focus:**

Sufficient demonstration on hardware

**Strengths And Weaknesses:**

** Strengths**
- Large volume of task/scene/object types and states.
- focusing on the tasks that the users want in their daily lives

**Weaknesses**
- Lack of explanations on 1)why adding the fancy object features (e.g., state transition, magnetic connection, etc.) was necessary and 2)how realistic the new features are. I want to hear something like ‘How will these features help us to solve important/difficult robotics problems in the real world' or ‘How did the sim2real gap decrease before and after adding this feature'.
- The large sim2real gap due to visuals makes this dataset less appealing. Regardless, I appreciate how the author explained the source of the sim2real gap in detail. At least showing the correlations between the simulation V.S. the real-world evaluation would help and, for example, showing the relationship 'RL-VMC < RL-Prim < RL-Prim.Hist' in simulation is maintained in the real world with a similar ratio, making this benchmark more reliable.

- Lack of realism on robot-object interactions. This is a dataset for “Mobile Manipulation” and I think this is a significant weakness of this dataset. (details are in the 'Issues' section)


**Summary Of Recommendation:**

I understand how realistic physics simulators are useful in Robotics, especially when learning RL policy and training/testing in large-scale (one or more rooms) environments. I agree with the author's motivation to build a human-centric benchmark based on a physics simulator for mobile manipulation.
However, BEHAVIOR-1K has a large sim2real gap, as the authors mentioned, which is a major reason why I am skeptical about this paper. Moreover, the unrealistic robot-object interactions captured in the supplementary videos are concerning. The issue can be a limitation of the omniverse itself; however, this does not change the fact that the dataset is less useful for the Robotics community targeting real-world applications. It is still questionable how much we can trust the result based on this benchmark.

---

> ### Author Response · Authors · 2022-08-25
> **Response to Reviewer Qfw6 (1/2)**
>
> **Comment:**
>
> Thank you for your constructive comments and helpful feedback.
> We appreciate your recognition of the large scale and diversity of activity/scene/object/states in our benchmark, as well as the motivation of focusing on tasks that users want in their daily lives.
>
> **Q1: Realism on robot-object interactions (grasping, penetration between the cloth and the gripper)**
>
> A1: We would like to clarify that our physics simulation is realistic for robot-object interactions. Our simulator supports two modes of grasping: physical grasping and simplified grasping. Physical grasping is fully simulated by the underlying PhysX 5 physics engine and strictly relies on friction between the gripper and the object for grasping, as in the real world. Simplified grasping artificially creates a rigid connection (“sticky-mitten”) between the gripper and the object when certain conditions are met, and is therefore less realistic.
>
> In our experimental evaluation, we adopted a pick primitive with simplified grasping to facilitate policy learning, given the challenging nature of our long-horizon household tasks. This is why the grasping shown in the supplemental video might not seem realistic. In the attached zip file, we included two new videos (“**physical_grasping_rigid.mp4**” and “**physical_grasping_cloth.mp4**”) that showcase fully-simulated physical grasping of a rigid object (bottle) and a soft cloth object (towel), where the grasping behavior is much more realistic because no simplifications or artificial grasping mechanisms are used.
>
> Thanks for pointing out the penetration between the cloth and the gripper. We realized that we accidentally turned off the collision between the two when recording the supplementary video. We have fixed the issue in the new video (“**physical_grasping_cloth.mp4**”), which shows more realistic physical interaction.
>
> We believe the high physical realism for grasping and low-level manipulation of OmniGibson will provide a useful tool for the robotics community that targets real-world applications to develop and evaluate new solutions.
>
> **Q2: Are the fancy object features (e.g. state transition, magnetic connection) necessary? How realistic are these new features? ‘How will these features help us to solve important/difficult robotics problems in the real world' or ‘How did the sim2real gap decrease before and after adding this feature?**
>
> A2: These features are essential. As shown in Figure 4, hundreds of activities require object features such as magnetic connection (e.g. hanging wallpaper, assembling furniture), state transitions (e.g. heating food - temperature,  wiping table - dirtiness level), and object composition and decomposition (e.g. making a strawberry smoothie). If we only simulate kinematics and dynamics, we will be restricted to purely pick-and-place activities, which consist of only about 20% of our activities.
>
> Most of the object features like fluid, flexible material, and thermal effects are realistic, as shown in our original supplementary video. We approximated state transitions and magnetic connections to balance simulation fidelity and speed. The principle we follow is that even with approximations, our simulator should still induce very realistic robot motion that would solve the task in the real world. For example, to clean a dirty table in BEHAVIOR-1K, the robot needs to grasp a soft towel, soak it with running water, and then physically touch each stain particle on the table surface. Although the soaking and cleaning mechanism isn’t 100% physically realistic, it’s a significant leap forward compared to other simulators that allow agents to turn something soaked or cleaned with a single discrete, symbolic action. Similarly, to create a strawberry smoothie in BEHAVIOR-1K, the robot needs to grasp strawberries and ice cubes, place them in a blender, close the blender, press a button, and finally pour the resulting smoothie (liquid) into a glass. The exact blending mechanism in the blender is simplified, but the required robot motion (placing the strawberry in the blender, pressing the button, etc) is very similar to that of the real world. As a result, we believe developing these object features helps us to broaden the diversity of activities that we can tackle in simulation, and more importantly, the robotics solutions developed in simulation have a high chance of being transferred to the real world because the robot motion is very similar.
>
>
> **Zip File:**
>
> /attachment/d5414000ce5c6b2f48d4962bc8e1da744144b4a1.zip

---

> > ### Author Response · Authors · 2022-08-25
> > **Response to Reviewer Qfw6 (2/2)**
> >
> > **Comment:**
> >
> > **Q3: The large sim2real gap due to visuals makes this dataset less appealing**
> >
> > A3: The apparent visual sim2real gap shown in the sim-real pair of first-person images in Figure 5 is because we didn’t perform any system identification of the camera parameters or lighting conditions in our simulator. We used the default camera setting of Omniverse and very roughly tuned the simulated light sources. In our sim-to-real experiment, we aimed to test zero-shot policy transfer without any bells and whistles. As a result, the simulated images look realistic and plausible, but different from the real images from the robot’s onboard camera.
> >
> > We conducted a new experiment to demonstrate that a minimal amount of system identification can significantly help close the visual sim-to-real gap. In OmniGibson we hand-tuned the lighting conditions, the reflectivity of the objects’ material, and the camera parameters, and re-generated a pair of images from the same viewpoint in simulation and in the real world. The two images are included in the attached zip file (called “**visual_gap_real.png**” and “**visual_gap_sim_after_sysid.png**”). They look much more similar than what is shown in Figure 5 (“**visual_gap_real.png**” and “**visual_gap_sim_before_sysid.png**”). This experiment shows that with the correct system identification, the visual sim2real gap can be dramatically reduced.
> >
> > To shed more light on the sensor sim-to-real gap, we additionally added Figure A.12 in the appendix of the revised paper that compares different sensor modalities between sim and real (with the original, pre-sysid setting, not the new one). We observe a smaller sim-to-real gap for the LiDAR sensor than for the RGB and Depth sensors. To alleviate this going forward, we plan to incorporate additional sensor noise models (e.g. Redwood Noise Model for PrimSense depth cameras [1] and various image augmentation methods for RGB) in the near future. Also, researchers and practitioners are welcome to apply existing sim-to-real techniques like domain randomization, domain adaptation, and system identification to further bridge the visual sim-to-real gap. Our sim-to-real experiment is only meant to be the first step towards sim-to-real transfer with BEHAVIOR-1K, and we sincerely invite the robotics community to join us in this journey.
> >
> > [1] Choi, Sungjoon, Qian-Yi Zhou, and Vladlen Koltun. "Robust reconstruction of indoor scenes." Proceedings of the IEEE Conference on Computer Vision and Pattern Recognition. 2015.
> >
> > **Q4: Show correlations between the simulation V.S. the real-world evaluation, e.g. showing the relationship 'RL-VMC < RL-Prim < RL-Prim.Hist' in simulation is maintained in the real world with a similar ratio**
> >
> > A4: Thanks for your suggestion! We fully agree that showing the correlations between the simulation v.s. the real-world evaluation helps to validate the simulator’s realism. Hence, we conducted additional sim-to-real evaluations of the RL-VMC policy and the RL-Prim policy, in addition to the RL-Prim.Hist. policy in the original submission. We observed a similar trend in the real world as in simulation: RL-VMC < RL-Prim < RL-Prim.Hist. RL-VMC still achieves zero success because of sparse reward and exploration difficulty. RL-Prim has worse performance than RL-Prim.Hist: on average, the robot with RL-Prim.Hist successfully places 0.64 cups/bottles, whereas the one with RL-Prim places 0. Qualitatively, RL-Prim. tends to get stuck in a repetitive action loop because the agent is unaware of its action history. This preliminary result offers us some confidence that OmniGibson can be a reliable testbed for future sim-to-real robotics research. We also included this result in Section G.2 of the appendix of the revised paper.
> >
> >
> > **Zip File:**
> >
> > /attachment/2f4e25e1a38d836412d0b174920601f260ea8cad.zip

---

> > > ### Author Response · Authors · 2022-08-27
> > > **Looking forward to the discussion**
> > >
> > > Dear Reviewer Qfw6,
> > >
> > > Thank you again for your constructive reviews, which have helped us improve the quality and clarity of the paper. We hope that we have been able to address your concerns and questions regarding the realism in robot-object interactions, the utility of object features of OmniGibson, the visual sim-to-real gap, and the reliability of our simulator as a sim-to-real test bed. As we approach the end of the rebuttal discussion period, please don't hesitate to let us know if you have any additional questions or comments. We look forward to the discussion!
> > >
> > > Thank you for your time.
> > >
> > > Warmest,
> > >
> > > Authors

---

### Official Review · Reviewer_5r6Z · 2022-08-05

**Originality:** Very Good
**Technical Quality:** Excellent
**Clarity Of Presentation:** Excellent
**Impact:** 4

**Recommendation:**

Strong Accept: I recommend accepting the paper and will argue for my recommendation even if other reviewers hold a different opinion.

**Summary:**

The presented work introduces a new simulation environment and a set of tasks commonly performed in household environments. It can be used to benchmark and develop robotic systems to perform such tasks. The dataset is significanly more complex than existing state of the art both in terms of appearance and dynamics realism and covers most desirable future usecases of household robotics. Authors also demonstrate example of sim-to-real transfer on a task demonstrating usefulness of the simulator to achieve real-world performance.

**Issues:**

I trust the authors did a good job on the software side and the code is fast, well documented, and easy to integrate (which is impossible to verify during review).
I would suggest the authors provide more details about the practical aspects of using the simulator:
1. License - is it free to use both for academic or also commercial use cases?
2. Computational requirements - is there a trade-off between fidelity / computation required / resulting performance?

**Quality Of The Limitations Section:**

Additional details required

**Reviewer Expertise:**

3: The reviewer is fairly confident that the evaluation is correct

**Robotics Focus:**

Sufficient demonstration on hardware

**Strengths And Weaknesses:**

Strengths:
1. Clear motivation - The selection of the provided tasks is clearly motivated based on user research. This focus on impact can focus the research community on the most desirable value proposition of household robotics.
2. Fidelity - Dataset surpasses previous datasets in terms of size, complexity, and dynamics realism.
3. Sim-to-real benchmark - Authors show that the dataset is useful for at least the task they demonstrated. The authors also analyze the limits of this sim-to-real transfer. Even though this evaluation is more of a qualitative nature rather than quantitative it provides examples to the community of how it can be used for other tasks.

Weaknesses:
1. Sim-to-real transfer - I believe the benchmark is a good environment for evaluating the algorithms and hopefully will contribute to the development of more efficient algorithms. I believe one of the key limitations of this work will be whether the policies trained in the simulator will be able to generalize well to the real world. The authors provide examples of this process on a sample task but to what degree this will work in general is yet to be answered.
2. Computation requirements - It is not clear how much compute is required to train useful policies on the benchmark. If the computation is too prohibitive it might limit the usefulness of the benchmark to less-funded researchers.
3. Quite possibly the use of scripted simulators is a dead-end in the progress of robotics systems and instead, simulators learned directly from the real world (for example by the use of NeRF-like techniques) will be the ones that are necessary to sufficiently bridge the sim-to-real gap unlocking practical results. That being said, scripted simulators are a useful tool in the meantime.

**Summary Of Recommendation:**

I am happy to see authors are providing this work as a new tool for the community to experiment with. For this reason, I recommend accepting the work. Ultimately, the impact of this work will be judged by its adoption by the community.

The only downside I could see is that the availability of simulators might take the focus of the community away from achieving practical results in the real world and methods that would allow that. Quite possibly the use of scripted simulators is a dead-end in the progress of robotics systems and instead, simulators learned directly from the real world (for example by the use of NeRF-like techniques) will be the ones that are necessary to sufficiently bridge the sim-to-real gap unlocking practical results. That being said, scripted simulators are a useful tool in the meantime.

---

> ### Author Response · Authors · 2022-08-25
> **Response to Reviewer 5r6Z (1/2)**
>
> **Comment:**
>
> Thank you for your constructive comments and helpful feedback.
> We appreciate your recognition of our paper’s motivation to focus on the most desired household activities judged by humans, our benchmark’s improvement of activity/scene/object/state diversity and scale over existing benchmarks, and our attempt to deploy a trained policy on a real robotic platform.
>
> **Q1: General feasibility of sim-to-real transfer**
>
> A1: We agree that the general feasibility of sim-to-real transfer for a robotics benchmark in simulation is essential for its success and longevity. In this submission, we were able to demonstrate that our trained policy in simulation can achieve partial success for a mobile manipulation pick-and-place task in the real world. We also attempted to shed some light on the main areas of sim-to-real gap. This is only our first step. Going forward, we will continue our efforts to rigorously test our benchmark’s sim-to-real potentiality in more activities, more robots, and more object states (e.g. wiping a table clean) and properties (e.g. cloth, fluid, etc). We sincerely invite the robotics community to join us in this journey.
>
> **Q2: Simulators learned directly from the real world (e.g. by the use of NeRF-like techniques) might be required to bridge the sim-to-real-gap?**
>
> A2: The idea of directly learning simulators from the real world using NeRF is very appealing. However, extending NeRF to dynamic scenes can be challenging. Many existing works only focus on temporal interpolation over one trajectory [1-3], which limits their applicability in robotics applications as they cannot handle different initial configurations and input action sequences. Recent works have also attempted to learn world models with NeRF/compositional NeRF [4,5], yet they typically focus on small-scale tabletop environments with limited numbers of objects. It is currently unclear how well they can scale to large-scale household environments with a much more diverse set of objects as we have in BEHAVIOR-1K. With this being said, we are very excited about incorporating NeRF-like techniques into BEHAVIOR-1K as a future exploration of bridging the sim-to-real gap.
>
> [1] Park, Keunhong, et al. "Nerfies: Deformable neural radiance fields." Proceedings of the IEEE/CVF International Conference on Computer Vision. 2021.
>
> [2] Du, Yilun, et al. "Neural radiance flow for 4d view synthesis and video processing." 2021 IEEE/CVF International Conference on Computer Vision (ICCV). IEEE Computer Society, 2021.
>
> [3] Li, Tianye, et al. "Neural 3D Video Synthesis From Multi-View Video." Proceedings of the IEEE/CVF Conference on Computer Vision and Pattern Recognition. 2022.
>
> [4] Li, Yunzhu, et al. "3d neural scene representations for visuomotor control." Conference on Robot Learning. PMLR, 2022.
>
> [5] Driess, Danny, et al. "Learning multi-object dynamics with compositional neural radiance fields." arXiv preprint arXiv:2202.11855 (2022).
>
> **Q3: Computational requirements for running the experiments in the paper**
>
> A3: For each run of our experiments, we use either a single Nvidia GeForce RTX 2080 Ti or a single Nvidia RTX A-6000 GPU, together with an Intel Xeon CPU, and 40GB of RAM. During training, the GPU memory usage is around 9GB. Depending on which task, the total training iterations range between 10k to 25k, and the total wall-clock training time range between 3.75 to 7.5 hours. We have included this information in Section F.3 in the appendix of the revised paper.
>
> **Zip File:**
>
> /attachment/6583b49cd092ed56e4de03be21ae6f64b1c248a6.zip

---

> > ### Author Response · Authors · 2022-08-25
> > **Response to Reviewer 5r6Z (2/2)**
> >
> > **Comment:**
> >
> > **Q4: Computation requirement for running the simulator in general. Is there a trade-off between fidelity / computation required / resulting performance?**
> >
> > A4: Yes, there is a trade-off between simulation fidelity and computation required / simulation speed. To evaluate the performance of OmniGibson under different conditions, we performed a rigorous speed test in two representative scenes \texttt{Rs\_int} and \texttt{restaurant\_hotel} with different numbers of objects, in a single-GPU, single-process setup. We adopt the “idle”” setup from iGibson 2.0 [6] and Habitat 2.0 [7]: a robot is placed in the scene (except the last row "- Robot"), and stays still with zero velocity action. At each time step, the simulator runs the physics simulation, extended object state update, and transition machine update loop, and renders a $128\times128$ RGB image. We use action time step of $t_a = \frac{1}{30}\text{s}$ and physics time step of $t_s = \frac{1}{120}\text{s}$ to be consistent with previous works. Our benchmark runs on a Ubuntu machine with Intel(R) Core(TM) i7-8700K CPU @ 3.70GHz and one Nvidia GeForce 2080 Ti GPU in a single process setting. The results are summarized in Table A.10 in the appendix of the revised paper. The table (named “**benchmarking_table.png**”) is also separately included in the attached zip file for convenience.
> >
> > OmniGibson runs at a comparable speed to previous works like iGibson 2.0 [6] under similar settings, but with much higher rendering quality thanks to ray-tracing. It maintains a reasonable speed even for a large scene with over 800 objects. OmniGibson is also highly configurable and provides a flexible interface for users to balance between simulation fidelity and speed given their use cases and research interests. If the user isn't interested in fluid or cloth in their tasks, they can turn off these features to harvest performance speedup. Similarly, if the user is only interested in kinematics-only rearrangement tasks, or non-robotics embodied AI applications (e.g. a virtual camera), they can turn off object state update, or remove the robot, respectively. We haven't conducted any performance optimization, and we are actively working on improving aspects that can provide immediate and significant speedups, e.g. in areas like object sleeping, mesh simplification, and more efficient object state update. Furthermore, since real-time ray-tracing and fluid/cloth simulation is still an active area of research, we expect the upcoming hardware and software advances from Nvidia will lead to significant performance improvements in these areas.
> >
> > [6] Li, Chengshu, et al. "igibson 2.0: Object-centric simulation for robot learning of everyday household tasks." arXiv preprint arXiv:2108.03272 (2021).
> >
> > [7] Szot, Andrew, et al. "Habitat 2.0: Training home assistants to rearrange their habitat." Advances in Neural Information Processing Systems 34 (2021): 251-266.
> >
> > **Q5: Licensing**
> >
> > A5: Our entire codebase will be open-sourced under the MIT license and carefully documented. Under the Individual license, NVIDIA offers creators and developers free access to Omniverse, which is the underlying platform that we build upon, for both academic and commercial use cases.
> >
> >
> > **Zip File:**
> >
> > /attachment/feb7a23b22093d47b4c12ee9cb9df7deaf6cf27c.zip

---

> > > ### Author Response · Authors · 2022-08-27
> > > **Looking forward to the discussion**
> > >
> > > Dear Reviewer 5r6Z,
> > >
> > > Thank you again for your constructive reviews, which have helped us improve the quality and clarity of the paper. We hope that we have been able to address your concerns and questions regarding the feasibility of sim-to-real transfer, comparison with simulators learned from the real-world, computation requirements, and licensing. As we approach the end of the rebuttal discussion period, please don't hesitate to let us know if you have any additional questions or comments. We look forward to the discussion!
> > >
> > > Thank you for your time.
> > >
> > > Warmest,
> > >
> > > Authors

---

### Meta-Review · Area_Chair_bnE5 · 2022-08-13

**Recommendation:** Accept (Oral)
**Confidence:** 4

**Metareview:**

Summary:

This paper proposes a comprehensive simulation benchmark named BEHAVIOR-1K, which covers 1,000 everyday activities grounded in 50 scenes with more than 3,000 objects annotated with physical and semantic properties. The OmniGibson simulator supports these activities via realistic physics simulation and rendering. Using these modules, the authors confirmed the initial study on transferring solutions learned with a mobile manipulator in a simulated apartment to its real-world counterpart.

Strength:

- The selection of the provided tasks is clearly motivated based on user research. This focus on impact can focus the research community on the most desirable value proposition of household robotics.
- The paper also deployed a trained policy in a real environment as part of the experimental section.
- Compared to existing benchmarks, behavior-1k seems to be a big leap forward given its sheer size in the number of activities, objects, scenes, and annotations.

Weakness:

- Lack of realism on robot-object interactions. As a dataset for “Mobile Manipulation”, this is a significant weakness of this dataset.
- For the sim2real transfer, it isn't easy to assess how physical object interactions skills can be transferred to the real world.
- Computation requirements - It is unclear how much computing is required to train useful policies on the benchmark.


**Best Paper Nomination:**

Yes

---

> ### Author Response · Authors · 2022-08-25
> **Response to AC and all reviewers**
>
> **Comment:**
>
> We wholeheartedly thank all the reviewers and the AC for their constructive comments and helpful feedback. We would like to summarize the progress we have made during the rebuttal phase.
>
> We have made clarifications in the rebuttal response and in the revised paper to address the following concerns and questions:
> - General feasibility of sim-to-real transfer of our benchmark
> - The motivation of developing features of non-rigid elements (e.g. fluid, cloth) and object features like state transitions and magnetic connection
> - The fidelity of these features and the prospect of using them to train policies
> - How to set up a task and randomize a scene
> - How to tune the physics of our simulator
> - Comparison to simulators learned directly from the real world (e.g. by using NeRF)
> - Computation requirements for running the experiments in the paper
> - Licensing
>
> We have conducted additional experiments to address the following concerns and questions. Most results have been included in the revised paper, and they are also additionally included in the attached zip file for reviewers’ convenience:
> - We provided two additional supplementary videos to showcase physically realistic low-level robot-object interaction (including grasping) of rigid and non-rigid objects, to demonstrate our benchmark can support research in both high-level task planning and low-level robot manipulation (called “**physical_grasping_rigid.mp4**” and “**physical_grasping_cloth.mp4**” in the attached zip file).
> - We conducted preliminary system identification to show our simulator’s potential of minimizing the visual sim-to-real gap with a new pair of sim-real images (called “**visual_gap_real.png**” and “**visual_gap_sim_after_sysid.png**” in the attached zip file).
> - We presented a new qualitative comparison between different sensor modalities in simulation and in the real world to illustrate the sensor sim-to-real gap (called “**sensor_comparison.png**” in the attached zip file).
> - We conducted a new round of sim-to-real experiments with RL-VMC and RL-Prim. to show the performance trend in simulation is preserved in the real world, shedding some light on the reliability of our benchmark (see Section G.2 in the revised paper).
> - We presented benchmarking results of our simulator under different conditions to illustrate the tradeoff between simulation fidelity and speed (called “**benchmarking_table.png**” in the attached zip file).
>
> We also updated Figure A.3-7 in the appendix of the revised paper to better visualize our scenes and objects. Most of our paper revisions are in the Appendix, included in the zip file. All revisions are shown in blue.
>
> We want to thank all the reviewers and the AC again for their time and efforts to help us improve the paper. We are excited to engage in further discussions during the rebuttal phase.
>
>
>
> **Zip File:**
>
> /attachment/f02fb454412fbf5d68dfe5a883a2539278bcaee1.zip